# Family Occurrence of an m.3303C>T Point Mutation in the *MT-TL1* Gene, Which Induces Cardiomyopathy Syndrome with/without Skeletal Muscle Myopathy

**DOI:** 10.3390/genes15101289

**Published:** 2024-09-30

**Authors:** Olga Fałek, Dorota Wesół-Kucharska, Ewa Starostecka, Dariusz Rokicki, Katarzyna Fortecka-Piestrzeniewicz, Łukasz Kępczyński, Dorota Piekutowska-Abramczuk, Elżbieta Ciara, Iwona Maroszyńska

**Affiliations:** 1Department of Intensive Care and Congenital Malformations of Newborns and Infants, Polish Mother‘s Memorial Hospital—Research Institute, 93-338 Lodz, Poland; katarzyna.fortecka-piestrzeniewicz@iczmp.edu.pl (K.F.-P.); iwona.maroszynska@iczmp.edu.pl (I.M.); 2Department of Pediatrics, Nutrition and Metabolic Diseases, Children’s Memorial Health Institute, 04-730 Warsaw, Poland; d.wesol-kucharska@ipczd.pl (D.W.-K.); d.rokicki@ipczd.pl (D.R.); 3Department of Endocrinology and Metabolic Diseases, Polish Mother‘s Memorial Hospital—Research Institute, 93-338 Lodz, Poland; ewastarostecka@wp.pl; 4Department of Genetics, Polish Mother‘s Memorial Hospital—Research Institute, 93-338 Lodz, Poland; lukasz.kepczynski@iczmp.edu.pl; 5Department of Medical Genetics, Children’s Memorial Health Institute, 04-730 Warsaw, Poland; d.abramczuk@ipczd.pl (D.P.-A.); e.ciara@ipczd.pl (E.C.)

**Keywords:** cardiomyopathy, lactic acidosis, mitochondrial disease

## Abstract

This paper discusses the cases of siblings that were born healthy, then diagnosed in their neonatal periods with cardiomyopathy and/or severe metabolic acidosis, which ran progressive courses and contributed to death in infancy. Molecular testing of the children confirmed the presence of an m.3303C>T point mutation in the mitochondrial DNA in the *MT-TL1* gene, which was also present in their oligosymptomatic mother and their mother’s sister, an asymptomatic carrier.

## 1. Introduction

Cardiomyopathy and metabolic acidosis manifesting in the newborn and early infant periods are diagnostic and therapeutic challenges for paediatricians/neonatologists. The most common causes are bacterial and viral congenital infections, including TORCH infections, and congenital heart defects. Congenital metabolic defects, including mitochondrial diseases, fatty acid β-oxidation disorders, and Pompe disease, are much less common causes. However, they are equally important, so they should be considered in differential diagnoses. 

Mitochondrial diseases (MDs) make up a clinically heterogeneous group of disorders that are caused by defects in the mitochondrial respiratory chain (MRC) and abnormalities in cellular oxidative phosphorylation (OXPHOS). Mitochondrial dysfunction leads to ATP deficiency, abnormal calcium metabolism, excessive production of reactive oxygen species (ROS), impaired apoptosis, and nitric oxide deficiency. Clinical manifestations of MDs mainly involve systems that require a lot of energy to function—the central nervous system (CNS), skeletal muscles, and sensory organs. However, isolated involvement of one organ is also possible (e.g., Leber optic nerve neuropathy) [1]. It is estimated that cardiomyopathy may occur in up to 40–60% of patients with MDs, and its detection after 6 months of age worsens the prognoses of MDs [2,3,4,5].

The *MT-TL1* gene plays a crucial role in mitochondrial functionality; it encodes mitochondrial tRNA for leucine, which is essential for proper translation of mitochondrial proteins involved in oxidative phosphorylation (OXPHOS). Pathogenic variants of *MT-TL1*, such as the m.3243A>G variant, disrupt this process, leading to defects in energy production. These disruptions can result in a range of mitochondrial disorders, including MELAS, which are characterised by a variety of clinical manifestations [6]. *MT-TL-1* is encoded by mitochondrial DNA (mt-DNA). mtDNA is inherited through the maternal line. The degree of heteroplasmy, i.e., the presence of mitochondria with both normal and damaged DNA in a cell, may be responsible for the various clinical pictures of carriers with the same variant [2,7]. 

Cardiomyopathy syndrome (CMS) with or without skeletal myopathy, which is conditioned by an m.3303C>T point mutation in mtDNA in the *MT-TL1* gene, can manifest with varying degrees of disease severity, ranging from severe cardiomyopathy, usually leading to death in infancy, to oligosymptomatic or asymptomatic courses, depending on the degree of heteroplasmy. CMS is a rare disease. Few reports are available in the professional literature [8,9,10]. 

The first reported case can be found in a study by Silvestri et al. (1994). The authors presented a family in which the proband and a sibling died of cardiomyopathy in infancy. They appeared to be carriers of a homoplasmic mtDNA point mutation. Family members, i.e., the mother’s relatives, were either asymptomatic or oligosymptomatic, and they were found to have a point mutation with a high degree of heteroplasmy [8].

The cause-and-effect relationship of the m.3303C>T variant of *MT-TL1* was also confirmed by Bruno et al. (1999). The researchers presented eight members of four unrelated families with this point mutation, who developed both severe cardiomyopathy in infancy and isolated skeletal myopathy syndrome. In one patient, skeletal myopathy and cardiomyopathy co-existed, but their symptoms of skeletal myopathy preceded their myocardial abnormalities by up to 10 years [9].

In addition, Palecek et al. (2012) described a case of a 35-year-old man with hypertrophic cardiomyopathy due to the m.3303C>T variant of *MT-TL1*, which was detected during routine post-ischaemic stroke diagnostic testing [10]. 

This paper discusses the cases of siblings that were diagnosed in their neonatal periods with cardiomyopathy and/or severe metabolic acidosis, which ran progressive courses.

## 2. Case 1

A 5-day-old female neonate was admitted to hospital because of reluctance to eat, vomiting, and anuria that had persisted for several hours. This girl was born spontaneously from a first pregnancy at 38 weeks of gestation with a birth weight of 2850 g, a good general condition, and an Apgar score of 10/10. The parents were unrelated, and the perinatal period was uneventful.

On admission to the ward, the girl was in average general condition, presenting features of dehydration: a parietal forehead below the level of the cephalic bone, dry skin, skin fold symptoms (+), bradycardia to 79/’ (with normal oxygen saturation of 98%), hypotension, hypothermia (her temperature was 35.7 °C), pale skin, a cyanotic triangle around the mouth, and a weak pulse in the femoral arteries. Blood was aspirated from her stomach. The girl was apathetic and exhibited weakened neonatal reflexes. Laboratory investigations showed extreme metabolic acidosis with hypocapnia (pH: 6.72, CO_2_: 23.8 mmHg, BE: 28.1 mmol/L), hyperlactataemia (86 mg/dL, normal level: 4.5–12.6) hypernatraemia, hyperosmolality, increased renal indices, hypertransaminasaemia, hyperuricaemia, hypertriglyceridaemia, and markedly elevated creatine kinase and lactate dehydrogenase.

The girl required mechanical ventilation, full parenteral nutrition, intensive hydration, pressor support, and, finally, renal replacement therapy due to failure to restore diuresis. On day 12, mechanical ventilation was discontinued, and on day 43, the girl was discharged in good general condition.

To obtain a detailed diagnosis, the patient underwent a number of additional investigations. Laboratory tests (C-reactive protein (CRP) and blood culture) ruled out sepsis, whereas an analysis of acylcarnitines by tandem mass spectrometry (TANDEM MS/MS), a urinary organic acid profile obtained using the GCMS method, and a plasma aminogram and transferrin isoforms (CDG) test ruled out congenital metabolic defects. The Bautler–Baluda test confirmed the absence of galactosaemia. The patient underwent echocardiography, which ruled out a congenital heart disease. A normal female karyotype of 46.XX was obtained in genetic tests.

One month after the first discharge from the hospital, the child again required hospitalisation. The girl was admitted to hospital urgently after an approximately one-minute incident of disturbed consciousness accompanied by upward turning of the eyeballs, which in turn was followed by complete whole-body flaccid paralysis. Upon admission, the patient did not demonstrate any neurological abnormalities, except for reduced vitality. A head computed tomography (CT) scan showed no abnormalities. Additional examinations confirmed metabolic acidosis with hyperlactataemia and hypocapnia of lesser degrees than during the previous hospitalisation. Diuresis was normal. Elevated transaminases and anaemia were notable. 

Infections with the hepatotropic viruses HBs, HCV, CMV, and EBV were excluded. On day 7, the girl was discharged in good general condition.

After another week, the patient was admitted to hospital with pneumonia (she presented with a cough, signs of pharyngitis, and low respiratory effort). Additional examinations again revealed metabolic acidosis with hyperlactataemia of an average degree, hypertransaminasaemia, and anaemia. The concentration of cardiac enzymes was notable, as it for the first time slightly exceeded the upper threshold of the reference value.

After one week, the girl was discharged in good general condition.

Two months later, the patient presented with symptoms similar to those observed during the first episode. They included extreme dehydration with features of circulatory centralisation and required immediate intervention in the paediatric intensive care unit with the application of mechanical ventilation. 

The only complaint reported by the mother was impaired appetite lasting for two days, with no diarrhoea, vomiting, or fever.

An analysis of the acid–base balance again showed metabolic acidosis with hypocapnia (pH: 7.17, BE: 17.7 mmol/L), extreme hyperlactataemia (166.6 mg/dL, normal values: 6.3–18.9), and plasma hyperosmolality. This time, the lactate concentrations were significantly higher than during previous hospitalisations. Hypertransaminasaemia and anaemia reoccurred. This time, the levels of cardiac enzymes were much higher than previously (CK-MB: 91 IU/L, normal values: <25; Troponin T: 81 pg/mL, normal values: <14), with an extremely high NT-proBNP value (>35,000 pg/mL, normal values: <125). Elevated creatine kinase levels and slight transient hyperammonaemia were also detected. 

Echocardiography showed features of hypertrophic cardiomyopathy in both ventricles with left ventricular predominance and mild atrioventricular valve regurgitation. Despite multidirectional measures, the child’s condition gradually deteriorated. On day 14, the child demonstrated symptoms of multiple organ failure and was pronounced dead. 

## 3. Case 2

A younger brother was born on time in good general condition with a body weight of 3400 g. The adaptation period was uneventful. 

In the second month of life, the boy was hospitalised in the Cardiology Department because of cardiac murmurs. An examination showed metabolic acidosis (pH: 7.29, SB: −10.2 mmol/L, HCO_3_: 13.1 mmol/L) and increased levels of lactic acid (11.8–7.5 mmol/L, normal level: <2.1). Echocardiography showed trace patent foramen ovale (PFO), patent ductus arteriosus (PDA), and a slightly hypertrophied ventricular septum (Table 1), with preserved normal contractility. An electrocardiogram (ECG) and a Holter ECG were normal. The boy’s development was normal; he was breastfed and gained weight.

At the age of 4 months, he was hospitalised again for congenital metabolic defects. At that time, the boy’s development was slightly delayed, mainly in the motor area. In addition, he was also diagnosed with hypotonia. In laboratory tests, metabolic acidosis with an elevated lactate level persisted. Additionally, elevated serum concentrations of pyruvic acid (2.7–3.4 mg/dL, normal level: 0.3–1.2) and alanine (858.5 µmol/L, normal level: 148–448) were found. The carnitine concentrations (free and total), biotinidase activity, and acylcarnitine profile in a dry blood drop were determined with the MSMS method and appeared to be normal, and the urinary organic acid profile, determined with the GCMS method, showed significant excretion of methylmalonic acid, 2-ketoglutaric acid, and lactic acid. Echocardiography was performed and showed progressive changes, with hypertrophy of both the ventricular septum and posterior wall. However, this did not narrow the outflow or inflow of either ventricle. Magnetic resonance imaging (MRI) results of the central nervous system obtained with the use of spectroscopy appeared to be normal.

Mitochondrial disease was suspected, treatment with riboflavin was started, and acidosis was compensated with oral natrium bicarbonicum.

At the age of 4.5 months, the boy experienced pneumonia, which manifested with features of circulatory insufficiency (increased effort during feeding, increased sweating, and decreased motor activity). After the introduction of antibiotics and propranolol, the boy’s condition stabilised. However, he did not demonstrate developmental progress, and in the following weeks, we observed regression, i.e., the boy stopped rolling over from his stomach to his back, and due to hypotonia, he could not hold his head. He still maintained eye contact, smiled, and chattered.

At 6 months, further progression of hypertrophic cardiomyopathy was observed (Table 1), with elevated NTproBNP and clinical features of circulatory failure.

Spironolactone (1.5 mg/kg/day) and lisinopril (0.06 mg/kg/day) were included in the treatment, but they did not significantly improve the patient’s condition, and he died a few weeks later.

In both case 1 and case 2, skeletal muscle myopathy was not diagnosed due to the absence of typical symptoms of this disease (lack of hypotonia and only a periodically elevated creatine kinase level). 

Nonetheless, cardiomyopathy without skeletal myopathy is still considered cardiomyopathy syndrome with/without skeletal muscle myopathy. 

## 4. Molecular Studies

For case 2, next-generation sequencing (NGS) was performed on a blood DNA sample on a HiSeq 1500 system (Illumina, San Diego, CA, USA) using the original CMHI NGS panel of 1000 clinically relevant genes (Roche Diagnostics, Rotkreuz, Switzerland). NGS was performed with a mean depth of 104×; the mean 20-fold coverage of the target and the mean 60-fold coverage of the mtDNA were 99%. The *ACAD9* gene was specified by a clinician. Then, the whole mitochondrial genome was screened for deleterious variants. Sanger sequencing was used to confirm detection of the m.3303T>C variant in the proband and his family members.

DNA isolated from peripheral blood revealed a known pathogenic variant (chrM:3303C>T, rs199474660) located in the *MT-TL1* mitochondrial leucine tRNA gene (*MT-TL1*:n.74C>T). The prevalence of this variant is significantly increased in affected individuals compared to controls (only one individual in population databases had this variant at heteroplasmy). The computational predictor MitoTIP (www.mitomap.org), an in silico tool dedicated to mitochondrial tRNA variants, confirmed this variant to be pathogenic (92.7 out of 100). The HmtVAR database, which collects mtDNA variants (https://ngdc.cncb.ac.cn/databasecommons/database/id/6172), allowed its pathogenic rating to be predicted (score: 0.65). Additionally, the Clinical Genome Resource (ClinGen; https://www.clinicalgenome.org/ accessed on 29 September 2024) expert group classified the m.3303C>T variant as likely pathogenic. Single-fibre testing showed higher levels of this variant in ragged red fibres that were COX (Cyclooxygenase)-negative (42.4 ± 7.0%) and ragged red fibres that were COX-positive (58.2 ± 5.8%), and they were significantly higher than the levels of this variant in fibres that appeared normal (10.7 ± 6.3%; ref). In summary, this variant meets the criteria to be classified as pathogenic for primary mitochondrial disease inherited in a mitochondrial manner. This variant was found to be almost homoplasmic (67/68 reads) via NGS. A post-mortem analysis conducted by direct Sanger sequencing confirmed that the same variant had a high degree of heteroplasmy in blood from the proband and his sister (case 1). Further familial evaluation revealed the same variant in their oligosymptomatic mother with a high degree of heteroplasmy as well as in the proband’s mother’s asymptomatic younger sister, who demonstrated a lower level of heteroplasmy (Figure 1).

Both assessments were performed on DNA isolated from urine samples. The level of heteroplasmy was arbitrarily estimated as high/low when comparing Sanger sequencing electropherograms. 

## 5. Discussion

The described cases of siblings with severe metabolic acidosis, hypertrophic cardiomyopathy, and skeletal myopathy (only case 2) are rare literary reports of cardiomyopathy syndromes with or without skeletal myopathy caused by an mtDNA variant in the *MT-TL1* gene. 

Extremely different phenotypes were observed in the presented cases of cardiomyopathy syndrome (CMS) with or without skeletal myopathy, starting with severe cardiomyopathy causing infant mortality and finishing with asymptomatic or oligosymptomatic courses. The latter patients are most often studied as the families of probands presenting with severe clinical symptoms [8,9,10]. The degree of heteroplasmy, i.e., co-occurrence of mitochondria with both normal and abnormal mtDNA in a cell, is one suggested reason for such a wide range of phenotypes correlating with one particular genotype. In carriers of a given mutation, the severity of clinical symptoms is greater if the degree of heteroplasmy is closer to homoplasmy, i.e., a condition in which all or almost all cellular mitochondria contain the mutated gene [7,10]. 

The varying levels of heteroplasmy are important factors in determining the clinical outcomes of mitochondrial disorders, including those caused by mutations in the *MT-TL1* gene. In these diseases, the proportion of mutated mtDNA within a cell significantly influences the severity and range of clinical symptoms. High levels of heteroplasmy, where the majority of mitochondria carry mutated mtDNA, often result in more severe manifestations. Conversely, individuals with lower levels of heteroplasmy may experience milder symptoms or remain asymptomatic, as the remaining healthy mitochondria can partially compensate for the dysfunctional ones. Despite this, the mechanisms underlying the clinical heterogeneity of mitochondrial disorders remain poorly understood. A post-mortem genotype–phenotype analysis of carriers of the common m.3243A>G mutation revealed additional mechanisms, such as correlations between high heteroplasmy and an overall diminished mtDNA copy number. Additionally, unexpectedly high levels of heteroplasmy (>75%) were observed in central nervous system samples that did not present the MELAS phenotype [11].

The medical histories of the two siblings presented here describe symptoms commonly seen in neonatal and infant intensive care units. However, the prevalence of mitochondrial diseases, especially cardiomyopathy syndrome with or without skeletal myopathy, means they are not suspected or diagnosed until the very end, after exclusion of congenital infections, including infections of the CNS and TORCH infections, and congenital malformations of the heart, urinary tract, gastrointestinal tract, and CNS. Therefore, after the above-mentioned pathologies have been excluded and the initiated management has appeared to be ineffective, it is worth checking whether this disease or another mitochondrial disease is a manifestation of a congenital metabolic defect. In the event of strong suspicion, one should consider performing a molecular test to confirm the diagnosis. It should be a next-generation sequencing test that takes into account various molecular causes of cardiomyopathy. 

Establishing a diagnosis definitely will not enable the initiation of a causal treatment. The therapy involves a standard treatment for cardiomyopathy and is hardly effective [2]. However, establishing a diagnosis is highly important for the family and the patient. It allows invasive, often painful diagnostic tests to be avoided, helps to establish a prognosis, allows the family to receive genetic counselling, and reduces costs associated with patient care.

## Figures and Tables

**Figure 1 genes-15-01289-f001:**
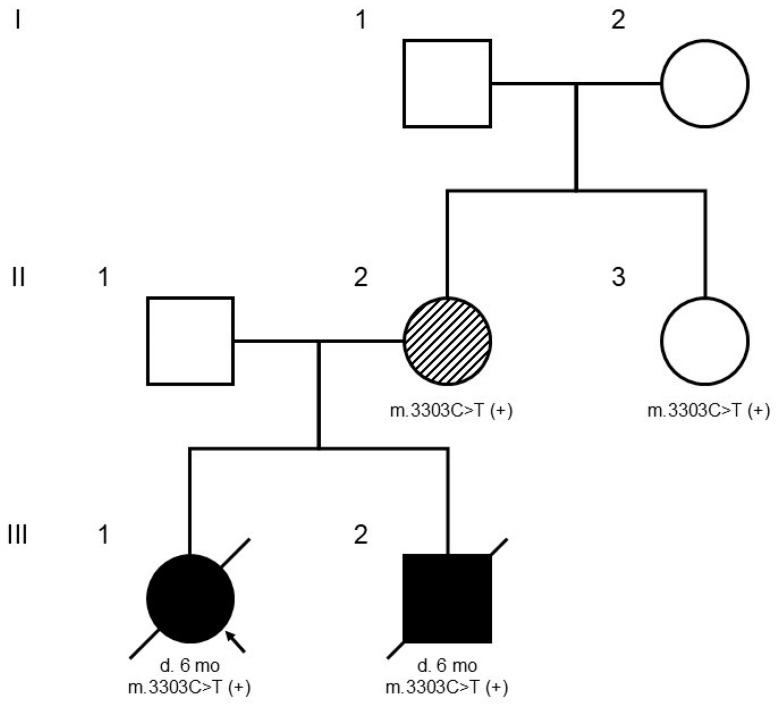
This figure presents the family pedigree, with an arrow marking the proband (III.1). The “m.3303C>T (+)” notation indicates individuals that were tested and confirmed to be positive for the mitochondrial m.3303C>T mutation, which is maternally inherited. The mother (II.2) is shaded to show her oligosymptomatic status, meaning she exhibited mild symptoms or few symptoms despite carrying the mutation. I, II, and III denote generations, 1, 2, and 3 denote individuals in generations.

**Table 1 genes-15-01289-t001:** Biochemical and cardiac parameters of patient 2 over time.

Age (Months)	2	4	5	6
BSA (m^2^)	0.33	0.34	0 34	0.36
IVSd (mm)	5.3	7.5	8.9	8.4
IVSd Z-score ^1^	+2.0	+5.1	+7.1	+6.4
LVPWd (mm)	5.9	7.6	8.6	10.2
LVPWd Z-score ^1^	+2.8	+5.4	+6.9	+9.4
RVDD (mm)	10.8	11.0	11.8	14.3
RVAWd Z-score ^1^	+0.9	+1.0	+1.4	+2.9
LV SF (%)	36	40	71	28
LV EF (%)	70	73	74	56
NTproBNP (pg/mL) normal value: 0–135	N/A	3440	2677	22,679–>35,000
Serum lactate (mmol/L) normal value: 0.3–2.2	7.5–11.8	9.0–14.5	8.3–10.5	6.8
CK MB (ng/mL) normal value: 0–6.6	9.9	N/A	13.0–16.2	15.2–24.6
CK (IU/L) normal value: 0–295	135	189	150	187

BSA—body surface area, CK—creatine kinase, CM MB—creatinine kinase muscle fraction, LV EF—left ventricular ejection fraction, IVSd—interventricular septal thickness, LVPWd—left ventricular posterior wall thickness, N/A—not applicable, NTproBNP—natriuretic peptide B, RVDD—right ventricular diastolic dimension, LV SF—left ventricular shortening fraction. ^1^ Kampmann C, Wiethoff C, Wenzel A, Stolz G, Betancor M, et al. Normal values of M mode echocardiographic measurements of more than 2000 healthy infants and children in central Europe. Heart. 2000; 83(6): 667–672.

## Data Availability

clinical data: https://basiw.mz.gov.pl/mapy-informacje/mapa-2022-2026/, accessed on 21 July 2024. Genetic data: -Data are not publicly available and will be available upon reasonable request from the corresponding author. The data are not publicly available due to privacy restrictions.

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
