# Peer review of "Family Occurrence of an m.3303C>T Point Mutation in the MT-TL1 Gene, Which Induces Cardiomyopathy Syndrome with/without Skeletal Muscle Myopathy"

_genes, 2024, doi:10.3390/genes15101289_

Round 1

Reviewer 1 Report

Comments and Suggestions for Authors

PDF attached

Comments on the Quality of English Language

English language editing is needed before publication. The manuscript needs to be edited for spelling, punctuation and grammar. There are also typos to be taken care of.

Author Response

Thank you very much for taking the time to review this manuscript. Please see the attachment (author-coverletter).

Should I give more details?  

Reviewer 2 Report

Comments and Suggestions for Authors

The work by Fałek et al., presents a case report of two siblings affected by a rare m.3303C>T variant in the MT-TL1 gene that caused a mitochondrial disorder. The study documents the detailed clinical progression, diagnostic and therapeutic challenges, and genetic findings associated with this variant and highlight the implications in cardiomyopathy and skeletal muscle myopathy that linked to the degree of heteroplasmy. Overall, this case report provides valuable insights into the clinical and genetic aspects of this rare mitochondrial disorder, which are clinically relevant and especially meaningful for the future diagnosis. I have the following suggestions:

1.       Please consider including a more structured and concise presentation of the clinical timeline and diagnostic steps. You may also consider using a flowchart to summarize the diagnostic pathway and key findings at each stage.

2.       The biochemical tables (Tables 1-4) are highly similar. You may consider condense and merge them for better visual clarity and compatibility.

3.       Please include details of NGS methodology, including the coverage, read depth, and validation steps.

4.       The authors may discuss how varying levels of heteroplasmy influence clinical outcomes in other cases of mitochondrial disorders or MT-TL1 mutations.

5.       Please indicate the unit for the X-axis in Figure 1 and revise the figure legend to include more technical details.

6.       The IRB and Informed Consent Statement at the end are missing.

Author Response

Thank you very much for taking the time to review this manuscript. Please see the attachment (Respond to the Reviewer 2).

Adding to the Comment 3:

Comment 3: Please include details of NGS methodology, including the coverage, read depth, and validation steps.

Respond 3: Thank you for taking the notice. Indeed, it was lacking the details about methodology. It's added in chapter: "Molecular studies" line: 187-194.

Adding to the Comment 4:

Comment 4: The authors may discuss how varying levels of heteroplasmy influence clinical outcomes in other cases of mitochondrial disorders or MT-TL1 mutations.

Respond 4: Thank you for your suggestion. We've proceded the new fragment to the "Discussion" part, line: 228-241.

The rest is included in attachment.

Reviewer 3 Report

Comments and Suggestions for Authors

This manuscript by Falek et al describes 2 siblings carrying a mutation in the MT-TL1 gene (m3303C>T), a mitochondrial encoded tRNA.

Abstract line 19: please change "a sibling" to siblings.

Please revise the introduction of this manuscript. It is written in a style that can be misleading and some of the information is not correct (see comment to line 46-47 below). For better understanding of the topic, some information on the importance of the gene MT-TL1 for mitochondrial functionality should be included.

Line 44: The authors write “The MT-TL1 gene is one those that conditions normal mitochondrial translation”. Please check, as this sentence appears incomplete.

Line 46-47: The authors write that the complexes of the respiratory chain are encoded by the mitochondrial DNA. This is not correct, as complex II is completely nuclear encoded and the remaining complexes consist of proteins encoded by the nuclear and mitochondrial genome.

The authors use several times the term “abnormal variant” when they address a point mutation (m.3303 C>T. Please change.

Line 55: Please change “ultra-rare” to rare, because “ultra-rare” begs the question when does a rare disease becomes an ultra-rare disease?

Table 1: Could it be that the reference values for ammonia and lactate dehydrogenase are switched?

Figure 1: For a non-medical mitochondrial researcher this figure is lacking details. Please provide more information. I don’t understand, what the word “fluid” or adrenalin means in context of these figures. Which kind of fluid?  Y-axis: Please increase the font size. What is the x-axis representing?

Line 115 states that the child was hospitalized again after 1 month. Please clarify, whether this is one month after birth or 1 month after the first discharge.

Line 157: a table 9 and a supplement is mentioned but is not included in this manuscript.

Molecular studies: it is not clear to me, whether the authors did these studies/experiments or how they obtained the results. Original data to these studies would improve this manuscript significantly. If these studies were done by the authors, please include protocols and present the data in more detail.

Line 197" COX-abbreviation is not defined. Are there differences between the siblings?

Since the girl showed symptoms earlier, it would be interesting to see, whether the mutation in the girl was closer to homoplasmy for the mutation than her brother. 

Considering the title of this manuscript, it is not clear which of the siblings had skeletal muscle myopathies and how these were diagnosed. Please add or change title.   

Author Response

Dear Editor, 

I'm very sorry, but I can't send you my corrected manuscript, cause genetician responsible for the main issue, that should be changed, is on his holiday till 31.08. I can't even contact him directly. I've received such message from his secretary. 

What shell I do then?

Best regards 

Olga Fałek 

Round 2

Reviewer 3 Report

Comments and Suggestions for Authors

Please keep table 3 in this manuscript, because a quantitative summary of the data of this table is given in the lines 148/149.

Author Response

Thank you very much for taking the time to review this manuscript. Please find the detailed responses below and the corresponding revisions/corrections highlighted/in track changes in the re-submitted files:

Comment 1: Ensure all references are relevant to the content of the manuscript.

Response 1: Agree. It was my mistake, indeed. I've provided the corrections: line 240,245.

Comment 2: Highlight any revisions to the manuscript, so editors and reviewers can see any changes made.

Response 2: OK. Done.

Comment 3: Provide a cover letter to respond to the reviewers’ comments and explain, point by point, the details of the manuscript revisions.

Response 3: Agree. I've sent 3 cover letters in a file.

Comment 4: If the reviewer(s) recommended references, critically analyze them to ensure that their inclusion would enhance your manuscript. If you believe these references are unnecessary, you should not include them.

Response 4: No, there was no references recommended.

Response 5: If you found it impossible to address certain comments in the review reports, include an explanation in your appeal.

Response 5: Thank you, everything went right.

Comment 6: Please keep table 3 in this manuscript, because a quantitative summary of the data of this table is given in the lines 148/149.

Response 6: Please kindly note, that there was no table 3 in these line. Did you mean "Table 1"? I've kept Table 1 in the manuscript.
